



# Multi-coupling analysis of temperate glacier stability: A case
# study of Midui glacier on Tibet, China
Guang Li [1,2], Minggao Tang [1,2] [*], Huanle Zhao [1,2], Daojing Guo [1,2], Xiaonan Yang [1,2], Xu Ran [1,2]
[1] State Key Laboratory of Geohazard Prevention and Geoenvironment Protection, Chengdu University
of Technology, Chengdu 610059, China
[2] College of Environment and Civil Engineering, Chengdu University of Technology, Chengdu 610059,
China
[*] Correspondence to: Minggao Tang (tmg@cdut.edu.cn)
**Abstract**:Temperate glaciers are particularly sensitive to climate warming. The instability of temperate
glaciers and their geohazards chains threaten the safety of residents and engineering facilities. However,
limited attention has been paid to the quantitative assessment of the stability of temperate glaciers, and
the response of dynamic characteristics and hydrothermal distribution to climate change is still unclear.
Herein, based on thermo-hydromechanical numerical simulation, the dynamic characteristics and
hydrothermal variation of temperate glaciers are analyzed, and a conceptual model for quantitative
evaluation of the stability and potential collapse area is proposed. The results show that: (1) The low
temperature area is mainly concentrated in the glacier upper reaches. The minimum temperature of the
glacier in the cold and warm season can reach -10 and -8 ℃, respectively, and the basal temperature is
maintained at melting point temperature. (2) The maximum flow velocity in the cold and warm seasons
are 45 and 50 m/yr, respectively. The maximum flow velocity is concentrated in the area with the largest
glacier thickness. (3) The glacier instability strip is located in the glacier upper reaches. During the year,
the factor of safety reached a maximum of 2.03 in February and a minimum of 1.48 in August.
## 1. Introduction
Glaciers are considered to be indicators of climate change. Temperate glaciers at low latitudes are
particularly sensitive to climate change (Li et al., 2008). According to the Blue Book on Climate Change
in China 2023, the global warming trend has progressed. From 1901 to 2022, the average surface
temperature in China increased by 0.16 ℃ every decade (China Meteorological Administration Climate



Change Center, 2023). As Asian water tower, the Tibetan Plateau (TP) has the fastest warming rate.
Rising temperatures have accelerated the melting and flow of mountain glaciers, leading to glacier
instability (Jacquemart et al., 2020). The hydrothermal changes and stability of glaciers have a profound
impact on the regional natural environment and social economy. The TP has become a hot spot for
studying climate and glacier changes (Wang et al., 2019).

In recent years, the glacial and periglacial environment is experiencing an increase in geohazards

due to cryosphere change (Faillettaz et al., 2015; Gilbert et al., 2015). The ice avalanches (IAs) caused
by glacier instability on the TP have increased. The sudden instability and geohazard chains of glaciers
will pose a serious threat to the safety of residents and engineering facilities downstream (Zhang et al.,
2023b). In 2016, two giant IAs occurred in the Ngari, Tibet, with a volume of $68 \times 10^6$ m$^3$ and $83 \times 10^6$
m$^3$, respectively, resulting in the deaths of 9 herdsmen and hundreds of livestock (Kääb et al., 2018). In
2018, the Sedongpu IA in southeastern Tibet caused an IA-debris flow-dammed lake (Li et al., 2022; Li
et al., 2024). There have been four IAs in Anyemaqen Mountains, Qinghai Province since 2004,
destroying large grasslands and affecting 732 herders (Zou et al., 2023; Zhang et al., 2023a).

Remote sensing images can observe glacier flow processes and area changes (Garg et al., 2019;

Millan et al., 2022; Altena et al., 2019). However, the lack of data in some remote sensing images will
affect the monitoring accuracy and resolution of time series (Cook et al., 2023). Glacier numerical
simulation has become the best method to analyze the dynamic process (Zhen et al., 2016). Through two-
dimensional and three-dimensional thermo-dynamic coupling models, the changes in the velocity and
stress field of the glacier are obtained to reveal the response mechanism to climate change (Zhang et al.,
2013; Wang et al., 2018; Zhao et al., 2013). The Full Stokes flow laws that comply with the mass and
momentum conservation is used to systematically describe the movement of glaciers. It can be combined
with shallow ice approximate model, shallow ice shelf approximate model, and high-order approximate
model. They are applied to the finite element numerical simulation, which can not only obtain the velocity
field of the glacier, but also the thermodynamic state (Zhao et al., 2022; Karlsson et al., 2021; Ai et al.,
2019). Based on the Full Stokes model, Seddik et al. (2019) studied the force balance state of the
Bowdoin glacier, and explained the influence of base lubrication and tide on the glacier stability. Gong
et al. (2017) evaluated the importance of basal boundary conditions in surging glacier transient



simulations. Glacier numerical simulations have been extensively studied. Due to the complex thermo-
hydromechanical coupling physical process of temperate glaciers, the fine simulation research on
temperate glaciers in southeastern Tibet is still scarce, especially the flow velocity, temperature, heat flux,
and stress analysis of glaciers under rising temperature. The stability of temperate glaciers is closely
related to thermodynamic processes. The research gap limits the accurate assessment of glacier stability.
Rising temperatures reduce the viscosity and shear strength of ice, and the increase of glacier flow
velocity makes the ice more prone to brittle fracture, thus inducing glacier instability. When the basal
fluid pressure at the ice-rock interface reaches a critical value, rapid sliding of the glacier will lead to
IAs. Regional differences in glacier change indicate that factors such as bedrock topography, elevation
distribution, subglacial meltwater, and moraine cover lead to different of the internal and external
dynamic coupling modes (Vallot et al., 2017; Jiskoot et al., 2017). Atmospheric temperature, rainfall,
and geological tectonic activity are key factors in studying the mechanism of glacier instability (Tang et
al., 2024). Faillettaz et al. (2015) pointed out that temperate glacier instability usually occurs when the
slope is about 30°. The rupture location occurs within the ice, on the bedrock, and in the bedrock in
temperate regions. Gilbert et al. (2016) analyzed the thermal status, basal friction and shear stress before
the Aru IAs, but did not quantitatively evaluate the glacier stability. Glacier instability often occurs at the
glacier scarp or periglacial region, and it is of great significance to clarify the stability changes and
potential collapse zone (Kääb et al., 2024).
Most mountain glaciers in low-altitude areas are temperate glaciers. The response of temperate
glacier dynamics and stability to climate change can provide some insights into the instability mechanism.
Herein, the Midui glacier is taken as an example. Based on the glacier thickness and meteorological data,
considering the multi-physical processes in the glacier flow, a thermo-hydromechanical coupling
numerical model is established to analyze the dynamic characteristics and hydrothermal changes of
glaciers under climate warming, and the glacier stability and potential collapse zone are evaluated.
**2. Study area**
The Midui glacier is located in Midui valley, Bomi County, Tibet (Fig. 1), and it is a typical low-
altitude temperate glacier. The Midui glacier presents an obvious firn basin, forming a very steep back





wall (Fig. 2a). The glacier fall is located under the firn basin, and it is at an altitude of 4850~4100 m.
The glacier tongue of Midui glacier is directly connected to the glacial lake (Fig. 2b), and the collapsed
glacier directly falls into the glacial lake (Fig. 2c).

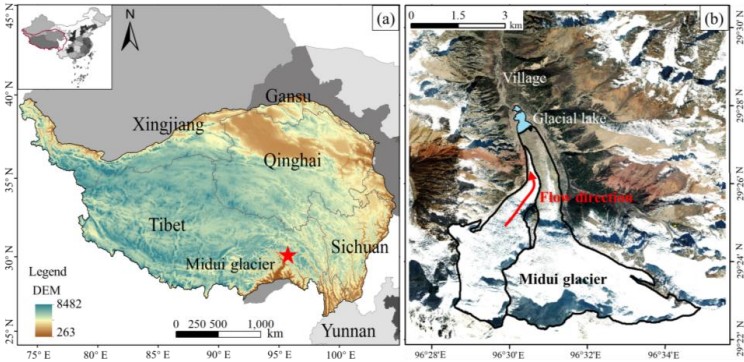


**Figure 1.** Geographical location of study area. (a) The Tibetan Plateau. (b) The Midui glacier.

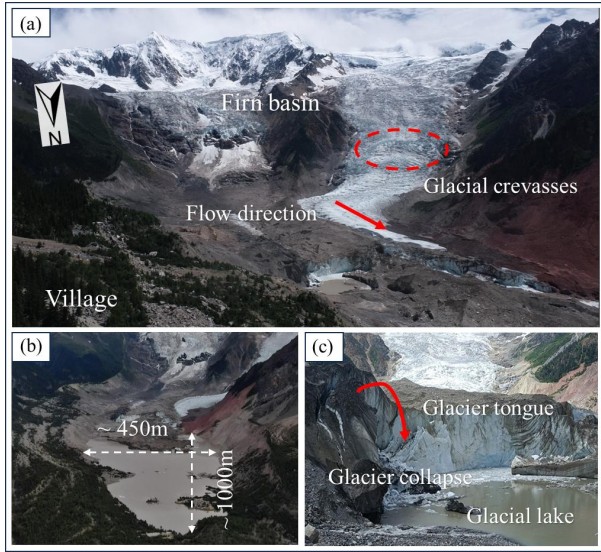


**Figure 2.** General introduction of study area. (a) The overview of study area. (b) The glacial lake. (c) The intersection
of Midui glacier and glacial lake.

The Glacier tongue is covered by moraine with a thickness of 50 ~ 80 cm (Fig. 3a and b). The main

lithologies are granite and limestone. There are many longitudinal and horizontal crevasses on the glacier
surface (Fig. 3c and d). The crevasses can be up to tens of meters long. The longitudinal crevasses are
usually wedge-shaped. The upper width is usually greater than 0.7 m, and the lower width is narrower.



It penetrates into glacier for several meters. Meltwater seeps into the glacier along crevasses and
recharges the glacial lake. There will be large rocks on the glacier surface. Compared with bare ice, the
melting rate of ice under large rocks is significantly reduced, resulting in the formation of glacial tables,
which are usually composed of a rock supported by narrow ice legs (Fig. 3e and f).

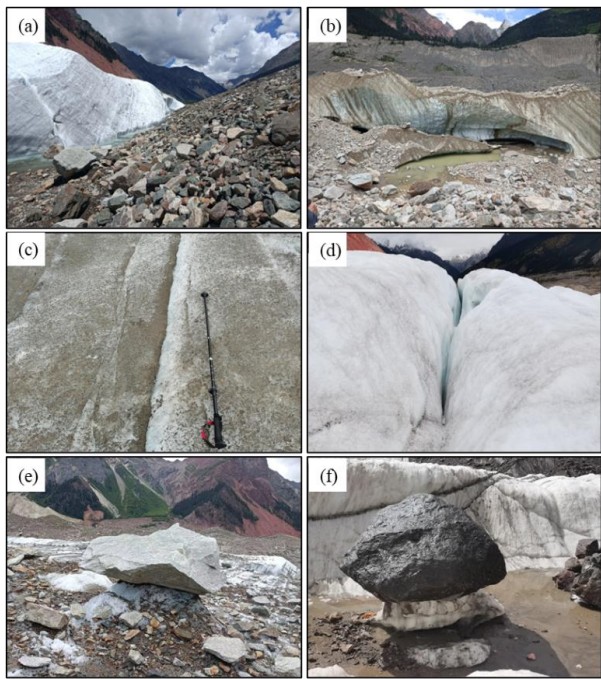


**Figure 3.** Geological phenomena during the flow of temperate glacier. (a) and (b) are glacial tills. (c) and (d) are
glacial crevasses. (e) and (f) are glacial tables.
On July 15, 1988, there was an IA in the study area. For meteorological-driven geohazards, the
meteorological data sets of 1 day (the day of the triggering), 7 days (6 days prior the triggering), 30 days
(29 days prior the triggering), and 90 days (89 days prior the triggering) can be analyzed to understand
the relationship between meteorological anomalies and geohazards (Paranunzio et al., 2024). Therefore,
we selected the average, minimum, and maximum temperatures ($T_{mean}$, $T_{min}$, and $T_{max}$) from April
17,1988 to July 15,1988 and the total precipitation (including rain and snowfall) from 1980 to 1988, as
shown in Fig. 4 (The data were acquired from Free Weather API: https://open-meteo.com/). There were
three wet years from 1980 to 1988: 1984, 1985 and 1988. The highest precipitation in 1988 (the year of
the triggering) was 2448.9 mm, which was 607.4 mm higher than that in 1987. In June 1988, the highest





temperature reached 30.1 ℃, close to the instantaneous maximum temperature for many years. In July
1988, the high temperature lasted for many days before IA, and the daily average temperature reached
7.6 ℃ ~ 11.6 ℃. The abnormally warm and humid climate caused a large amount of meltwater to seep
into the glacier base. The buoyancy and seepage pressure of the glacier tongue increased, and the
frictional resistance of the glacier base decreased, resulting in an IA with a scale of $3.6 \times 10^5$ m$^3$ (Dang et
al., 2019). The disintegrated glacier poured into the glacial lake to form glacier lake outburst floods
(GLOFs).

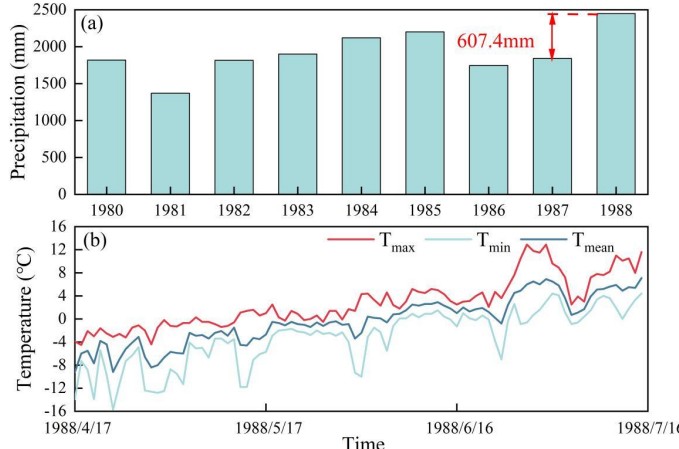


**Figure 4.** Historical climate analysis. (a) The total precipitation (including rain and snowfall). (b) The average,
minimum, and maximum temperatures.
**3. Methods**
**3.1. Mathematical model**

There have been serious IA and geohazard chain in the study area. In order to explore the impact of

climate change on glacier stability and temperature field, a conceptual model was used to describe it (Fig.
5). The momentum and continuity equations and incompressible heat equations have been used to
describe the ice velocity, ice temperature, and stress distribution. Subglacial water flow occurs within the
permeable layer and is considered as unsaturated seepage, which is described by the Richards equation.
The specific equation descriptions are as follows.



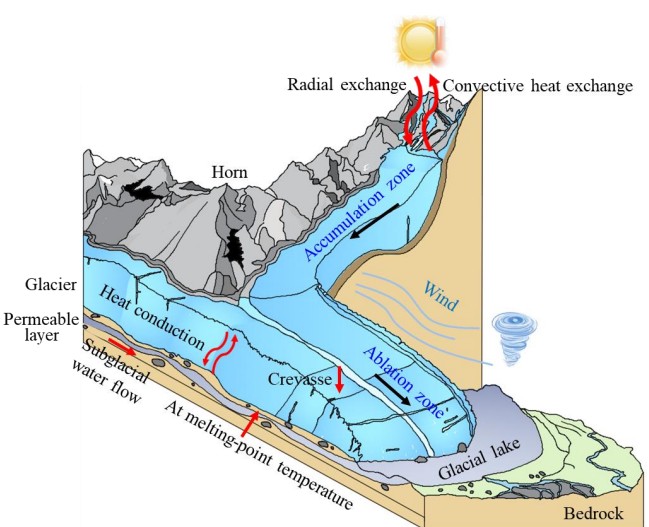

**Figure 5.** A conceptual model akin to the Midui glacier.

### 3.1.1. The temperature equation

Glacier flow controls the basic process of ice material deformation and energy budget. The

temperature distribution of glacier can be calculated by energy balance equation (Tannehill et al., 1997).
$$\rho c_p \frac{\partial T}{\partial t} + \rho c_p u \cdot \nabla T = Q + \nabla (k \cdot \nabla T) \tag{1}$$

Here, $c_p$ is the heat capacity; $T$ is the temperature; $u$ is the velocity vector; $Q$ is the heat source
term; and $k$ is the thermal conductivity of ice. The heat capacity and thermal conductivity of ice change
with temperature (Ritz, 1987), expressed as:
$$c_p (T) = 146.3 + 7.253T \tag{2}$$

$$k (T) = 9.828 e^{-0.0057T} \tag{3}$$

Herein, the heat source term considered the convective heat exchange ( $q_c$ ) and radiative heat

exchange ( $q_r$ ) between the ambient temperature and the glacier surface. The convective heat exchange
uses forced convective heat flux (Incropera et al., 2007):
$$-n \cdot q_c = h (T_{ext} - T) \tag{4}$$



$$h = \begin{cases} 2\dfrac{K_{ext}}{L}\dfrac{0.3387\,\mathrm{Pr}^{1/3}\bullet\mathrm{Re}^{1/2}}{\left[1+\left(\dfrac{0.0468}{\mathrm{Pr}}\right)^{2/3}\right]^{1/4}} & \mathrm{Re} \le 5\times10^{5} \\[4ex] 2\dfrac{K_{ext}}{L}\,\mathrm{Pr}^{1/3}\left(0.037\,\mathrm{Re}^{4/5}-871\right) & \mathrm{Re} > 5\times10^{5} \end{cases}$$
(5)

Here, n is the outward unit normal vector; h is the convective heat transfer coefficient; $T_{ext}$ is the
ambient temperature; L is the characteristic length; $K_{ext}$ is the air heat transfer coefficient; $\mathrm{Pr}$ is the
Prandtl number.

Radiative heat transfer is based on the Stefan-Boltzmann law:

$-\mathrm{n}\bullet q_r = \sigma\varepsilon_s\left(T_{ext}^4 - T^4\right)$
(6)

Here, $\sigma$ is the Stefan-Boltzmann constant; $\varepsilon_s$ is the land surface emissivity.

The temperature of the bedrock can be assumed to be the melting point temperature (Wagner et al.,

1994):

$T = T_{tp} - b_{CC}\left(p - p_{tp}\right)$
(7)

Here, $T_{tp}$ is the melting point temperature of water; $p_{tp}$ is the standard atmospheric pressure; $\beta_{CC}$ is
the Clausius-Clapeyron constant.

**3.1.2. Subglacial water flow**

There is a large amount of meltwater at the temperate glacier base. The subglacial water flow plays

a role in transporting the fine particles produced by the glacier erosion. The water flow is accompanied
by a large number of debris and soil to form unsaturated seepage, which is described by Richards equation
(Richard, 1931):
$\rho\left(\dfrac{C_m}{\rho g}+S_e S\right)\dfrac{\partial P}{\partial t} + \nabla\rho\left(-\dfrac{K_s}{\mu}K_r\left(\nabla P + \rho\mathrm{g}\nabla D\right)\right) = Q_m$
(8)

Here, $C_m$ is the bulk density of water; $\rho$ is the water density; $S_e$ is the effective saturation:
$S_e = \dfrac{(\theta - \theta_r)}{\theta_s - \theta_r}$; $\theta$, $\theta_s$, and $\theta_r$ are the water content, saturated water content, and residual water content,



respectively; S is the storativity; $P$ is the pressure; $K_s$ is the saturated hydraulic conductivity; $\mu$ is
the fluid viscosity; $K_r$ is the relative permeability; D is the position head; $Q_m$ is the source item.
When the soil-water characteristic curve, the relationship between soil moisture content and
permeability coefficient, and the corresponding initial values and boundary conditions are known, the
unsaturated seepage problem can be analyzed based on the Richards equation. The van Genuchten model
is used to describe the hydraulic characteristics of the subglacial permeable layer (Van, 1980).
$$h = -(S_e^{-\frac{1}{m}} - 1)^{\frac{1}{n}} / \alpha \tag{9}$$
$$S_e = \frac{\theta_l - \theta_r}{\theta_s - \theta_r} = \begin{cases} \dfrac{1}{\left[ 1 + |\alpha h|^n \right]^m} & h < 0 \\ 1 & h \geq 0 \end{cases} \tag{10}$$
The relationship between pressure head and liquid water content is obtained by combining the above
equations:
$$h = -\left[ \left( \frac{\theta_l - \theta_r}{\theta_s - \theta_r} \right)^{-\frac{1}{m}} - 1 \right]^{\frac{1}{n}} / \alpha \tag{11}$$
$$K = \begin{cases} K_s S_e^l \left[ 1 - (1 - S_e^{\frac{1}{m}})^m \right]^2 & h < 0 \\ K_s & h \geq 0 \end{cases} \tag{12}$$
Here, $h$ is the pressure head; $m$, $n$, and $l$ are empirical parameters, and $m = 1 - \dfrac{1}{n}$; $\alpha$ is the the
reciprocal of soil air entry value; $\theta_l$ is the liquid water content; $K_s$ is the saturated hydraulic
conductivity; $K$ is the hydraulic conductivity.
**3.1.3. The velocity equation**
Motion is the inherent natural property of glaciers. As an incompressible non-Newtonian fluid, the
motion equation of glacier can be expressed by Stokes equation, which can reflect the basic mechanical
law of viscous fluid flow and satisfy the conservation of momentum (Tannehill et al., 1997):
$$\rho_i \frac{\partial \mathbf{u}}{\partial t} = \nabla \bullet [-pI + \mu(\nabla u^T)] + \rho_i g \tag{13}$$



$\nabla \bullet u = 0$     (14)
Here, $\rho$ is the ice density; u is the velocity vector; p is the pressure; g is the acceleration of gravity;
$\mu$ is the dynamic viscosity.

Ice shows a change in viscosity when it is sheared. The relationship between shear strain rate and

shear stress of ice can be expressed by Glenn flow law (Glen, 1955):
$\mu = \dfrac{1}{2} A^{-\frac{1}{n}} \bullet \gamma^{n-1}$     (15)
Here, $\gamma$ is the shear rate, which can be defined as the strain rate tensor modulus:
$\gamma(u) = \left\| \dfrac{1}{2}\left(\nabla u + \nabla u^{T}\right) \right\|$; n is the flow characteristic index of ice; A is the flow rate factor.

To fully define the Glenn flow law, it is also necessary to define the consistency index of ice:

$m = \dfrac{1}{2} A^{-\frac{1}{n}}$     (16)
Here, m is the consistency index of ice.

The viscosity of ice depends not only on the shear rate, but also on temperature and pressure. The

Arrhenius law is used to represent the flow rate factor, which is the expression of temperature (T) and
pressure (p) (Stocker et al., 2013):
$A(T, p) = A_{0} e^{\left(\frac{-Q}{RT}\right)}$     (17)
Here, $A_{0}$ is the flow rate constant, and the expression is $A_{0} = \begin{cases} 3.985e^{-13} & T \le -10\text{℃} \\ 1.916e^{3} & T > -10\text{℃} \end{cases}$ ; Q is the
activation energy, and the expression is $Q = \begin{cases} 60e^{3} & T \le -10\text{℃} \\ 139e^{3} & T > -10\text{℃} \end{cases}$ ; R is the Stefan-Boltzman.
**3.1.4. Mechanical equations**

There are shear fracture surfaces caused by crevasses before the glacier instability, which will cause

rapid disintegration and collapse of glacier under gravity. The strength reduction method has been proved
to be reasonable for predicting the glaciers stability lacking large displacement precursor information
(Wei et al., 2024). Therefore, elastic-plastic constitutive equation and strength reduction method were
used to study the glacier stability. The glacier stability can be considered as a mechanical problem



involving matrix failure. The quantitative evaluation of glacier stability can refer to the slope stability
evaluation method, and the elastic-plastic analysis adopts Mohr-Coulomb criterion. The Mohr-Coulomb
yield function and the related plastic potential are:
$$F = Q = \mathrm{m}(\theta)\sqrt{J_2} + \alpha I_1 - k \tag{18}$$
$$\mathrm{m}(\theta) = \sqrt{\frac{1}{3}\left((1+\sin\varphi)\cos\theta - (1-\sin\varphi)*\cos\left(\theta + \frac{2\pi}{3}\right)\right)} \tag{19}$$
$$\alpha = \frac{\sin\varphi}{3} \tag{20}$$
$$k = c \cdot \cos\varphi \tag{21}$$
Here, $\theta$ is the Lode angle; $J_2$ is the second invariant of deviatoric stress tensor; $I_1$ is the first
invariant of stress tensor; c is the cohesion; $\varphi$ is the internal friction angle.
The cohesion and internal friction angle of glacier ice are set as a quadratic function related to
temperature. The function is obtained by the author 's laboratory test (Tang et al., 2023). Their
relationships with temperature are as follows:
$$c = 155.04 - 1.49T - 0.02T^2 \tag{22}$$
$$\varphi = 1.91 - 1.23T - 0.02T^2 \tag{23}$$
The strength reduction method can reflect the instability process of glaciers after strength
degradation. This method is to reduce the shear strength parameters in equal proportion until the glacier
reaches the limit equilibrium state. The reduction coefficient at this time is the factor of safety (FOS):
$$c' = \frac{c}{FOS} \tag{24}$$
$$\varphi' = \arctan\left(\frac{\tan\varphi}{FOS}\right) \tag{25}$$
Here, $FOS$ is the reduction coefficient; $c'$ and $\varphi'$ are the reduced cohesion and internal friction
angle, respectively.
**3.2. Computational model and parameters**
(1) Numerical model
In order to analyze the velocity, temperature and stress distribution of the Midui glacier, a two-
dimensional numerical simulation was established. Firstly, the section line is drawn based on the





231 centerline of the glacier movement (Fig. 6a). As shown in Fig. 6b and c, the topography of the surface

232 and base is obtained from the surface elevation and ice thickness data (The DEM data were acquired

233 from: https://earthexplorer.usgs.gov/, and the glacier thickness were acquired from:

234 https://www.research-collection.ethz.ch/handle/20.500.11850/315707). The base topography is the

235 surface elevation minus the ice thickness. Finally, the two-dimensional cross-section of the Midui glacier

236 is obtained (Fig. 6d). The length of the glacier computational domain is 874.7 m, and the altitude

237 difference is 730 m. The thickest of glacier reaches 77.9 m, and the thinnest is 9.37 m. The thickness of

238 the subglacial water flow permeability layer was assumed to be 1 m.

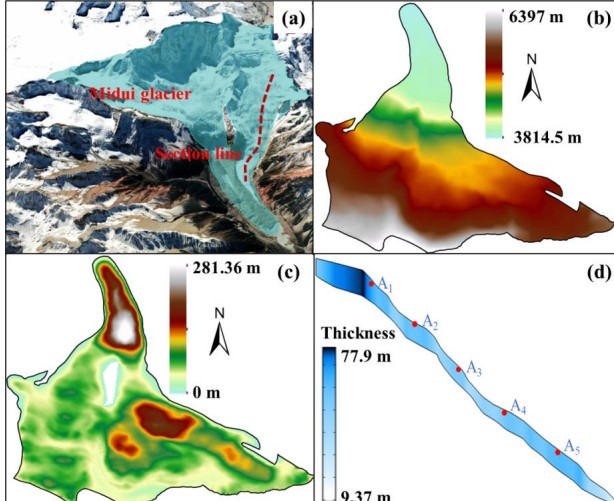

**Figure 6.** Computational model for glacier. (a) Longitudinal section diagram. (b) Glacier DEM. (c) Glacier thicknes.
(d) The 2D section of computational model.

(2) Boundary conditions and parameter

243  For heat transfer, the glacier base is set to the melting point temperature. The left and right

244 boundaries are set as free boundaries, and the heat flux can enter and leave the domain. The glacier

245 surface undergoes convective heat exchange and radiative heat exchange with the atmospheric

246 environment, where the wind speed is set to the average value of 1.01 m/s over the past 20 years. The

247 transient temperature field boundary condition of the glacier surface adopts the Dirichlet boundary, which

248 is obtained by directly giving the boundary temperature value, that is, a variable about space and time is

249 known:





$T = T\left(x, y, t\right)$     (26)
Here, we use the boundary layer theory to fit the long-term observed atmospheric temperature
function with the temperature increment. The empirical formula of glacier surface temperature change
can be obtained.
The seepage field is used to analyze the subglacial permeable layer. The left of the permeable layer
is a symmetrical boundary with a certain height of water head. The bottom is an impervious boundary,
the right is a seepage boundary, and the top is a non-flow boundary.
For the velocity field, the basal temperature is at the melting temperature, and the glacier can slide
through the bottom. The ice-rock interface is set to the viscous slip boundary condition. The glacier
surface is a stress-free boundary. The left and right boundaries are normal constraints. The left boundary
exerts pressure on the glacier, thereby accelerating the glacier flow, and the right boundary generates
resistance to the flow.
The force field is used for stability analysis of glaciers. First, the stress field of the model under
gravity is calculated, and the stress is provided to the model as prestress for recalculation, thereby
achieving stress balance and making the model have initial stress but no initial strain. The glacier surface
and the right boundary are free boundary, the glacier base is a fixed constraint, and the left boundary is
a horizontal constraint with vertical displacement freedom.
According to the references of the above formula, the parameters of each physical field in the
calculation model are shown in Table 1.
**Table 1.** Model parameters

| Parameter | Value (unit) | Description |
|---|---|---|
| $\rho_i$ | 910 (kg/m³) | Density of ice |
| E | $5.4 \times 10^9$ (Pa) | Young's modulus |
| $\nu$ | 0.35 | Poisson ratio |
| $\varepsilon_s$ | 0.97 | Land surface emissivity |
| n | 3 | Flow characteristic index of ice |
| $\mu_i$ | $5 \times 10^{12} \left( Pa \cdot s \right)$ | Dynamic viscosity of ice |
| $\beta_{cc}$ | $9.8 \times 10^{-8} \left( K / Pa \right)$ | Clausius-Clapeyron constant |



| $T_{tp}$ | 0.01 (℃) | Melting point temperature of water |
|---|---|---|
| $p_{tp}$ | 611.657 (Pa) | Standard atmospheric pressure |
| R | 8.314 $J/(mol \bullet K)$ | Universal gas constant |
| $\sigma$ | $5.67 \times 10^{-8} \left[ W / \left( m^2 \bullet k^4 \right) \right]$ | Stefan-Boltzmann constant |
| L | 874.7 (m) | Characteristic length |
| $\theta_s$ | 0.2 ( $m^3 \cdot m^{-3}$ ) | Saturated water content |
| $\theta_r$ | 0.05 ( $m^3 \cdot m^{-3}$ ) | Residual water content |
| $K_s$ | $1.66 \times 10^{-3}$ ( $m \cdot s^{-1}$ ) | Saturated hydraulic conductivity |
| m | 0.5 | Van Genuchten model parameter |
| l | 0.5 | Van Genuchten model parameter |
| $\alpha$ | 1 ( $m^{-1}$ ) | reciprocal of soil air entry value |
| $T_m$ | -4.41 (℃) | Annual average temperature |

**4 Results**

**4.1 Model verification**

In order to verify the reliability of the numerical coupling model and calculation method, we monitored the temperature at 1.2 meters below the glacier surface and glacier displacement in the study area (Fig. 7a, b). Field monitoring showed that the minimum temperature at 1.2 m below the ice surface was about-1.0 ℃, which occurred in mid-January. After 5 months, the glacier displacement reached 4.64 m. By comparing the observed values with simulated values (Fig. 7c, d), it can be found that they have the same change trend and good consistency. The root mean square error (RMSE) and mean absolute percentage error (MAPE) of temperature are 0.21 ℃ and 20.2%, respectively. The RMSE and MAPE of displacement are 0.14 m and 5.53%, respectively. Therefore, the model and numerical simulation results are basically reliable.

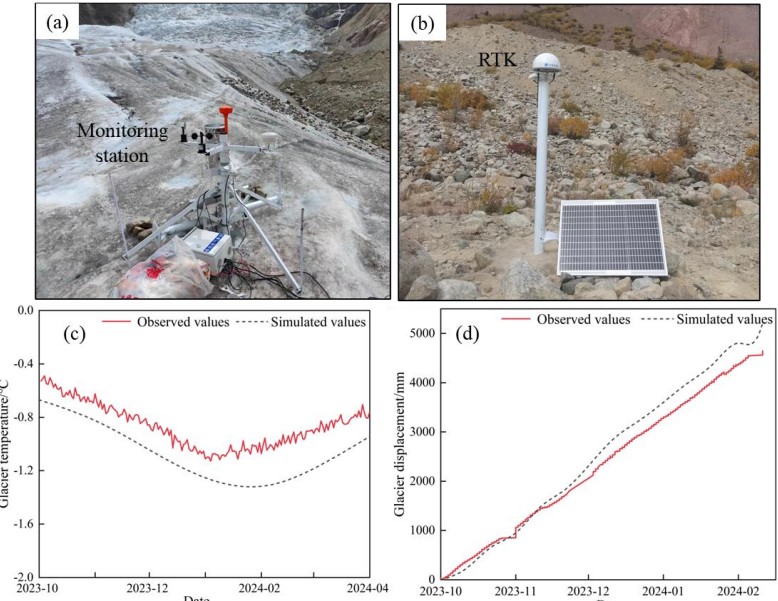

**Figure 7.** Comparison of observed and simulated values. (a) and (b) The monitoring station. (c) The glacier temperature. (d) The glacier displacement.

## 4.2 Glacier temperature field characteristic

Glacier temperature is controlled by surface and base temperatures and internal hydrothermal exchange processes, and it has a great influence on the glacier dynamics. Fig. 8 shows the glacier temperature distribution in cold and warm seasons. We observed that the glacier temperature in the cold season is significantly lower than that in the warm season, and the low temperature area is mainly concentrated in the glacier upper reaches. In the cold season, the minimum temperature reached -10 ℃, and the maximum temperature is lower than the ice melting temperature, indicating that the glacier is in material accumulation. The basal temperature is generally higher than the glacier surface temperature due to geothermal energy. In the warm season, the minimum temperature reached -8 ℃. At this time, the surface temperature is the highest, followed by the basal temperature, and the temperature in the glacier layer is the lowest. It indicates that the air temperature has the greatest contribution to the heat of the glacier.

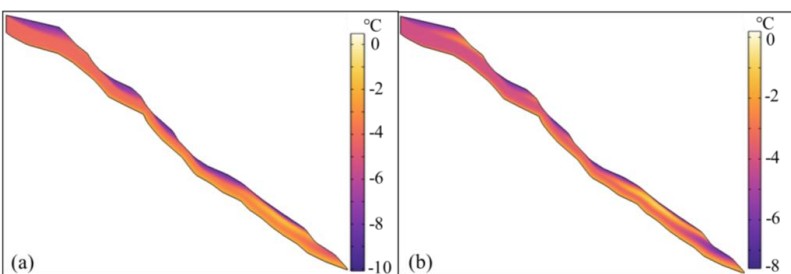

**Figure 8.** Glacier temperature distribution. (a) The cold season. (b) The warm season.

Fig. 9 shows the temperature variation of the glacier surface and base in January and July, respectively. It can be seen that the surface temperature is more sensitive to the external environment temperature. The glacier surface temperature varies greatly between January and July. The maximum fluctuation range is 9.1 °C. There is almost no difference in the basal temperature between January and July, which is maintained at melting point temperature, and the fluctuation range is not more than 0.25 °C. Seasonal temperature fluctuations on the glacier surface have little effect on the temperature of the glacier base, and the heat source is derived from the geothermal heat.

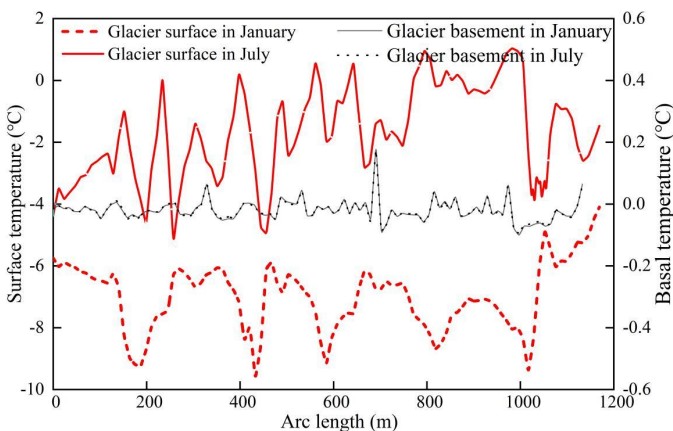

**Figure 9.** Glacier surface and base temperature changes.

The energy exchange between the glacier surface and the atmosphere affects the freezing and melting process, which can be further characterized by the heat flux. Fig. 10 is the heat flux change of the selected five points on the glacier surface in the past 10 years. The coordinate points are: $A_1$ (179.89, 4825.3), $A_2$ (321.95, 4699.5), $A_3$ (427.55, 4574.5), $A_4$ (568.5, 4449.8), and $A_5$ (726.02, 4325) (Fig. 6d). The positive heat flux indicates that the glacier surface is absorbing heat and melting. The negative heat





flux indicates that the glacier surface is exothermic and frozen, which is beneficial to the material balance
of glaciers. It can be seen that the higher the glacier altitude, the greater the fluctuation range of heat flux.
With the altitude increases, the air density decreases, and the rarefied air has lower efficiency in heat
conduction and convection, so the heat is more likely to accumulate or dissipate quickly. At the same
time, glaciers are often covered by snow at high altitudes, which increases the albedo of the sun and
reduces the absorption of solar radiation. Due to the loss of long-wave radiation at night, the temperature
difference between day and night increases and the fluctuation of heat flux increases.
During the year, the glacier surface absorbs more heat than it releases, indicating that the glacier
material balance is in deficit and the glacier is retreating. In the first year, the heat flux was positive in
late April, indicating that the glacier began to melt from late April, until October when the heat flux
became negative and the glacier began to remain stable or thicken. By the 10th year, the glacier began to
melt in early April. It shows that as the temperature rises over the years, the melting of the glacier will
gradually increase. The material balance of the glacier is at a loss, and the loss will gradually increase.

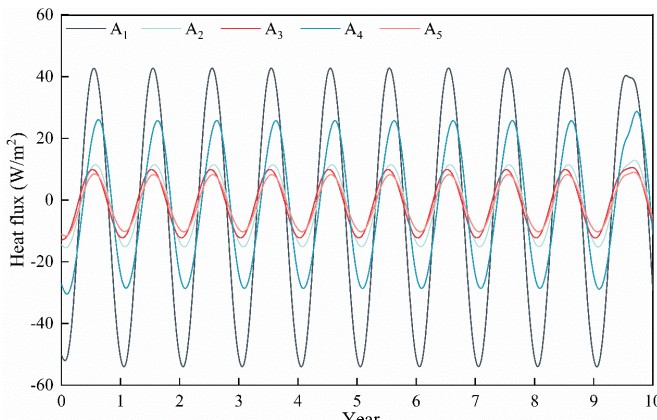


**Figure 10.** Glacier surface heat flux changes.
**4.3 Glacier flow velocity and outward mass flow rate**
Ice is a viscoelastic material that deforms under gravity. Its flow is determined by both the internal
properties of the glacier and the external environment. Fig. 11 shows the flow velocity of glaciers in the
cold and warm seasons after 10 years. The glacier velocity in the warm season is significantly greater
than that in the cold season. The maximum flow velocity in the cold and warm seasons are 45 and 50



m/yr, respectively. Because it is a temperate glacier, there is water lubrication between the glacier and
bedrock, and the base slides significantly. The maximum flow velocity is concentrated in the area with
the largest glacier thickness. The thicker glaciers will produce greater basal pressure, increasing the rate
of base sliding and glacier deformation.

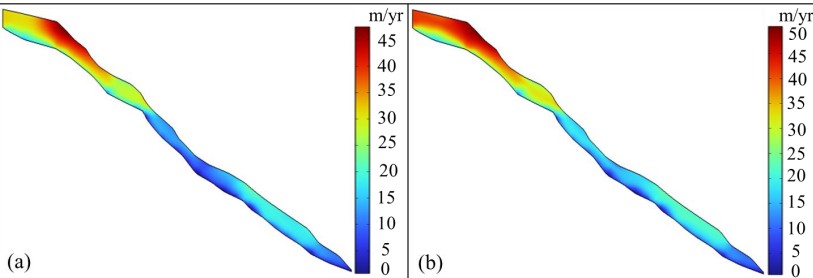

**Figure 11.** Glacier flow velocity distribution. (a) The cold season. (b) The warm season.

Mass flow rate is an important index to measure the mass balance of glaciers. The positive mass

flow rate indicates that the glacier is melting. The outward mass flow rate of the glacier is controlled by
glacier dynamics, physical properties, and environmental conditions. The variation trend of the outward
mass flow rate of the glacier in 10 years is shown in Fig. 12. It can be seen that the mass flow rate
increases year by year, and the growth rate suddenly increased in the sixth year, indicating that the glacier
is melting more and more violently. In the 10th year, the outward mass flow reached $6.13 \times 10^{-2}$ kg/s. The
increase of outward mass flow accelerated the glacier flow, and glacier crevasses were generated. Glacier
collapse and IA are more likely to occur, posing a threat to surrounding areas such as Midui village.

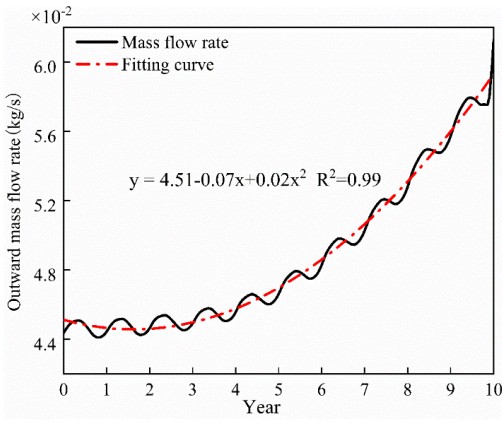

**Figure 12.** Outward mass flow rate of glacier.




**4.4 Glacier stability analysis**

Glacier stability analysis is to study the instability behavior and potential risk of IAs under different conditions. Herein, according to the thermo-hydromechanical coupling model and strength reduction method, the FOS of glacier can be obtained. Fig. 13 shows the relationship between the maximum displacement and the reduction coefficient of the glacier in January. At the beginning of the curve, the maximum displacement has remained at about 15 cm with the increase of reduction coefficient, and the displacement is small. When the reduction coefficient increases from 1.96 to 1.97, the maximum displacement in the glacier changes abruptly, increasing to nearly 1400 cm, and the curve has an obvious inflection point. Therefore, the FOS is preliminarily determined to be 1.96.

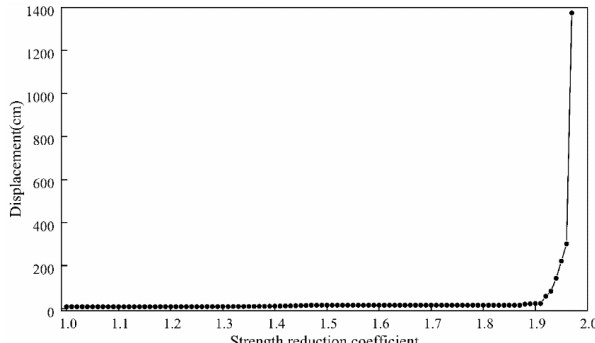

**Figure 13.** Variation of the glacier maximum displacement with reduction coefficient.

In order to further determine whether the glacier has reached the critical slip state when the reduction coefficient was 1.96, the plastic strain and the maximum shear strain were shown (Fig. 14). The maximum shear strain can be characterized by tresca stress, which refers to the difference between the maximum normal stress and the minimum normal stress. It can be seen that when the reduction coefficient is 1.96, the plastic zone at the glacier terminus has been completely penetrated and is arc-shaped. The maximum plastic strain is $3.1\times10^{-3}$. The tresca stress reaches the maximum value of $7.4\times10^5$ N/m$^2$ at the same position, and the area with the highest shear stress is the most dangerous. According to the instability criterion, the FOS of glacier is 1.96.





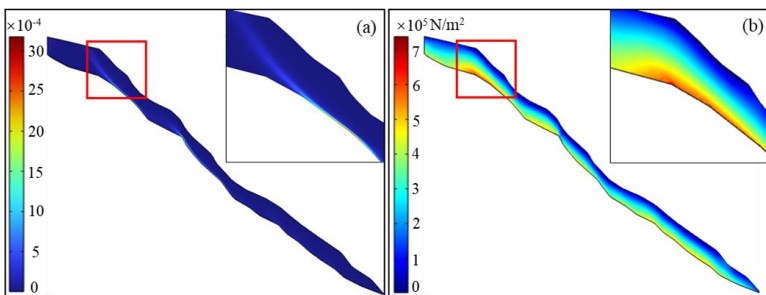


**Figure 14.** Plastic zone (a) and tresca stress (b) of glacier when the reduction coefficient is 1.96.
Fig. 15 shows the glacier displacement distribution when the reduction coefficient is 1.97. It can be
seen that the glacier instability strip is mainly located in the glacier terminus. On the one hand, the slope
on the glacier terminus is the steepest, and the distribution of crevasses is concentrated. Glacier is more
prone to collapse. On the other hand, the glacier thickness in this region is the largest. The larger the
thickness, the greater the basal pressure caused by gravity, which affects the basal sliding and glacier
stability.

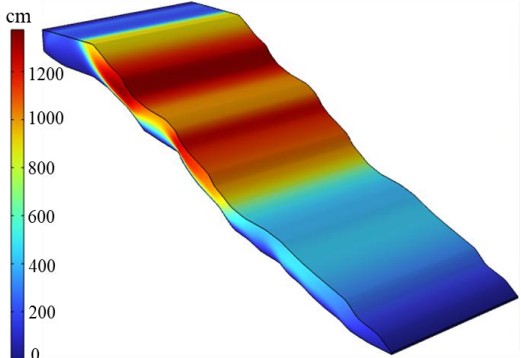


**Figure 15.** Glacier displacement when the reduction coefficient is 1.97.
Through the unsaturated seepage simulation of the permeable layer, the subglacial fluid pressure is
obtained when the reduction coefficient is 1.96 (Fig. 16). The fluid pressure at the inlet and outlet of the
glacier is low, and reaches the maximum in the upstream region. The high fluid pressure may cause the
basal sliding, resulting in more prone to IA in this area. There are two peaks of fluid pressure in the
process of decreasing, which is caused by the change of slope.



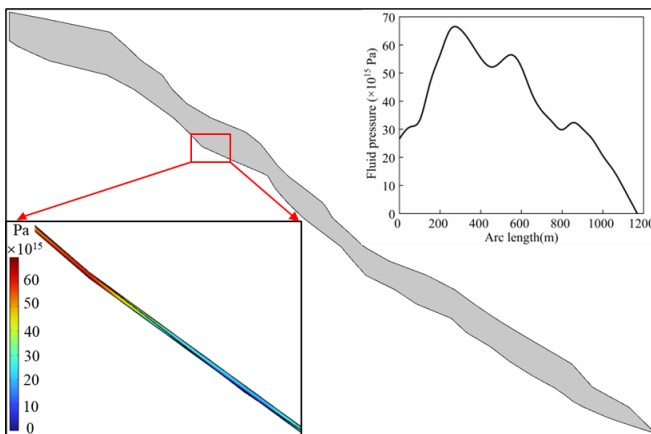

**Figure 16.** Subglacial fluid pressure.

In order to analyze the influence of monthly temperature change on glacier stability and maximum displacement, the monthly FOS and maximum displacement in the model are extracted (Fig. 17). With the change of monthly temperature, there is an obvious response relationship between FOS and maximum displacement. In the cold season, the FOS is large, the glacier stability is high, and the displacement of IA is also small. With the arrival of the warm season, rising temperatures reduce the glaciers stability and increase IA displacement. In February, the FOS reached a maximum value of 2.03, and the maximum displacement was only 1332 mm. The FOS decreased the most from May to July, and reached the minimum value of 1.48 in August. Since September, with the arrival of cold season, the FOS has gradually increased.

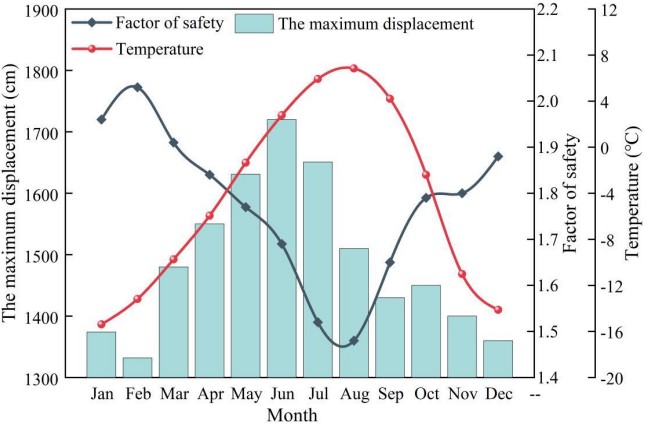

**Figure 17.** Changes in glacier maximum displacement and factor of safety.





**5. Discussion**

Through statistical analysis, there have been 62 recorded IAs on the TP since the 20th century. After the 21st century, the TP shows significant warm and humid characteristics, and the number of IAs has reached more than 20 times (Tang et al., 2024). These IAs are mainly distributed in southeastern Tibet. The glaciers in this area are relatively low in altitude, and most of them belong to temperate glaciers. The stability evaluation and instability mechanism analysis of temperate glaciers are particularly important. The glacier instability mechanisms induced by rising temperature, rainfall and earthquake are described below (Fig. 18).

The rising temperature reduces the temperate glacier stability through melting, lubrication and reduction of ice mechanical properties. The temperature difference causes thermal stress on the glacier surface, which promotes the expansion and deepening of crevasses. Meltwater infiltrates along the crevasses, and the increase of seepage pressure further enhances the crevasses expansion. With glacier ablation and crevasses expansion, the stress concentration area of glaciers is more likely to induce IA. When the meltwater penetrates into the glacier base, a lubricating layer is formed, resulting in a decrease in friction of the ice-rock interface and an increase in the subglacial water pressure. It makes the glacier accelerate the basal sliding and induce IA. For temperate glaciers, the basal temperature is the melting point, the ice viscosity decreases, and the plastic flow accelerates. Rising temperatures will also reduce the shear and tensile strength of ice, making it more susceptible to crack and collapse within the glacier.

Rainfall can penetrate into the glacier through crevasses, expanding existing crevasses. The rainfall increases the water pressure in the crevasses, causing the crevasses to expand deeper. At the same time, more meltwater generated by rainfall infiltrating into glacier base will also produce a lubrication effect. There are multiple tension cracks at periglacial region, and heavy rainfall will increase the glacier weight, causing slip breaking type IAs at periglacial region.

After the earthquake, the glacier will redistribute internal and external stress. The earthquake can directly induce IAs in a short time. It may also have a lag effect, causing subsequent IAs for a long time. The earthquake will aggravate the expansion and penetration of the glacier crevasses. The liquid water in the crevasses reduces the energy dissipation caused by the seismic wave, and the vibration is easier to



spread to the glacier base. Seismic waves may lead to a sudden increase in the shear force between the
glacier and base, and also cause local heat accumulation in the glacier, which induces IAs.

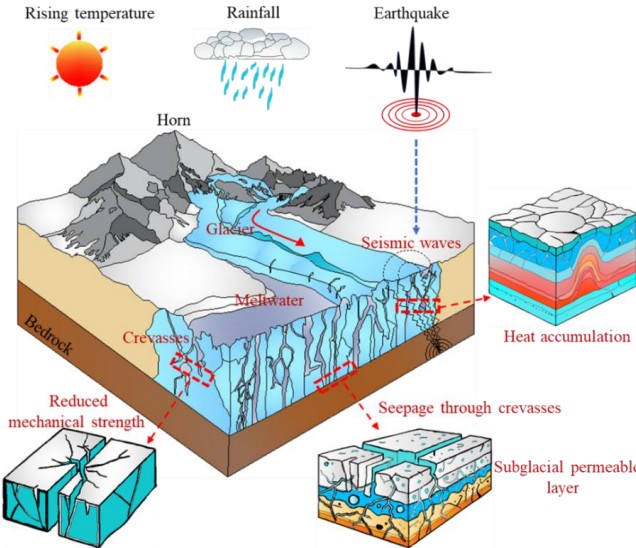


**Figure 18.** Temperature, rainfall and earthquake induced ice avalanche.
**6. Conclusions**
Herein, considering the multi-physical factors and their coupling mechanism in the glacier flow
process, the dynamic characteristics and hydrothermal distribution of temperate glaciers with climate
change were studied, and a conceptual model for quantitative evaluation of the stability and potential
collapse area was proposed. The field monitoring and simulation results are compared to verify the
reliability of the numerical coupling model and calculation method. The following conclusions were
drawn:
(1) The low temperature area is mainly concentrated in the glacier upper reaches. The minimum
temperature of the glacier in the cold and warm season can reach -10 and -8℃, respectively. The surface
temperature is sensitive to the external environment temperature. The basal temperature is maintained at
melting point temperature, and the fluctuation range is not more than 0.25 ℃. The higher the glacier
altitude, the greater the fluctuation range of heat flux. The total heat flux in one year is positive, and the
time when the heat flux becomes negative is gradually advanced with the increase of years.



(2) The glacier flow velocity in the warm season is significantly greater than that in the cold season.
The maximum flow velocity in the cold and warm seasons are 45 and 50 m/yr, respectively. The
maximum flow velocity is concentrated in the area with the largest glacier thickness. The mass flow rate
of the Midui glacier increases year by year. In the 10th year, the outward mass flow reached $6.13\times10^{-2}$
kg/s.
(3) The abrupt change of displacement, the penetration of plastic strain, and the generalized shear
strain are used as the criterion of glacier instability. The glacier instability strip is arc-shaped and located
in the upper reaches, where the fluid pressure in the subglacial permeable layer is the highest. In February,
the FOS reached a maximum of 2.03. The FOS decreased the most from May to July, and reached the
minimum of 1.48 in August./

*Data availability*. Data will be made available on request.

*Author contributions*. **Guang Li:** Investigation, Methodology, Software, Formal Analysis, Validation,
Writing – original draft, Writing – review and editing. **Minggao Tang:** Conceptualization, Methodology,
Data curation, Funding acquisition, Supervision, Resources, Writing – review and editing. **Huanle Zhao:**
Investigation, Methodology, Writing – review and editing. **Daojing Guo:** Writing – review and editing.
**Xiaonan Yang:** Investigation. **Xu Ran:** Investigation.

*Competing interest*. The authors declare that they have no conflict of interest.

*Acknowledgments*. This research has been supported by the National Natural Science Foundation of
China (Grant No. 42377199), Chengdu University of Technology Postgraduate Innovative Cultivation
Program (Grant No. CDUT2023BJCX008), the Second Tibetan Plateau Scientific Expedition and
Research Program (STEP) (Grant No. 2019QZKK0201), and State Key Laboratory of Geohazard
Prevention and Geoenvironment Protection Independent Research Project (Grant No. SKLGP2021Z005).



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

3.