# Peer review of "Multi-coupling analysis of temperate glacier stability: A case study of Midui glacier on Tibet, China"

_EGUsphere, 2025_

## Referee Comment (RC1)

The manuscript entitled "*Multi-coupling analysis of temperate glacier stability: A case study of Midui glacier on Tibet, China*" by Li et al. characterizes hydrothermal and ice dynamic characteristics of a temperate glacier on the Tibetan Plateau. The manuscript discusses glacier stability and geohazards associated with temperate glacier collapse, and states that the objective of the study is to quantitatively assess temperate glacier stability. The authors present model results for an along-flow cross section of the Midui Glacier. These model results represent ice temperature and velocity for warm and cold seasons. Ice mass flux for a ten-year period is evaluated. The authors quantitatively assess the factor of safety (FOS) for the studied glacier.

However, the results and analysis presented in the manuscript do not directly address glacier stability. This needs to be clarified in the title, abstract, and introduction. The 4.4. Glacier stability analysis section (L350-351) vaguely states that "according to the thermo-hydromechanical coupling model and strength reduction method, the FOS of glacier can be obtained." However, the FOS is never defined. Therefore, it is not clear how this metric relates to (a) model output, nor (b) glacier collapse. Furthermore, I find it misleading to say that a conceptual model is developed herein, when previously developed models are applied to a case study glacier.

I recommend that this manuscript be reconsidered after major revisions to address this and the following comments:

General and major comments

Insufficient technical detail for glacier modeling.

- Technical details for model methods are confusing. For example, section 3.1.1. The temperature equation presents an energy balance equation which includes a temperature term, presumably the temperature of the glacier ice. But is it the temperature of the glacier at the surface? The bed? How does the temperature of the bedrock (Equation 7) enter this modeling of ice temperature?
- Establish context for modeling choices. Subglacial water flow section needs to reference other work that uses this modeling approach, more recent than Richard (1931).
- The model spatial and time domains are unclear. Eventually, the reader can understand that the model results are for a cross-section (2D) spatial domain of the study glacier, but this – and the time domain – are not clearly communicated in the model methods section. For example, Figure 12 shows "Year 0-10" on the x-axis but what are the corresponding calendar years?

- How does the modeled stress field evolve? Line 262-263 "First, the stress field of the model under gravity is calculated" yet gravity is always the driving force for the glacier. This condition does not change. The geometry (surface slope, ice thickness) can change the driving stress, but does the model evolve these terms?

Insufficient technical detail for input data.

- The DEM used to define the study glacier surface is from earthexplorer.usgs.gov but this citation is too general. What was the method of DEM development? What is DEM uncertainty?

Figure captions are terse. Elaborate.

- Figure 7 could explain the instruments shown in each monitoring station.
- Figure 9 does not state the different y-axis scales for surface temperature.
- Figure 10 does not explain A_1, A_2, etc.
- Figure 16 does not explain the two inset plots and the main plot has no axis labels.
- Vertical exaggeration on each cross section needs to be defined.

Limited model validation.

- Ice velocity is validated using observational field data collected at a debris-covered ice location? It is confusing that the RTK "monitoring station" photograph (Figure 7b) depicts debris and vegetation. Where is the ice margin? Where is this on the study glacier? I am skeptical that validating ice velocity at this location informs the validity of modeled ice velocities at different locations, such as the upper glacier.

Unclear connection between physical processes discussed vs. studied:

- For example, Line 316 mentions albedo effects in the snow-covered accumulation zone on glacier heat flux yet this is the only time "albedo" is mentioned in the paper. Thus, it is unclear if the model takes this (potentially important) feedback effect into account.
- Another example is L334, which mentions basal water pressure, but it is unclear whether the model results account for this variable?

Define the Factor of Safety (FOS).

- FOS is first presented in L351, then FOS results are given in L356, without any context on how this metric is generated.
- Is this metric novel? Has it been used in other studies? Explain.

Discussion is theoretical and does not invoke results from the study.

- For example, the opening section of the discussion addresses earthquake effects on glacier stability, whereas the study does not address, model, or mention earthquakes.
- The Discussion is short (4 paragraphs) and does not explain how the results of the study enhance our understanding of ice collapse.
- Authors might consider discussing study results in the context of research summarized in the Introduction.

Line item and minor comments:

Capitalize "Midui Glacier" throughout. I flagged the first few, but not all, instances.

Change "material balance" to "mass balance" throughout.

Change "outward mass flow" to "mass flux" to throughout.

Define "cold season" and "warm season" and explain how these seasons are determined.

Distinguish catastrophic ice avalanche (glacier collapse) from lake calving. For example, Figure 2 labels the lake-terminating glacier front as "glacier collapse" yet how is this different from lake-calving ice?

Define "geohazard chain" and clarify which specific hazards it encompasses.

Line 27. Provide a citation to support the statement, "... the Tibetan Plateu (TP) has the fastest warming rate."

Line 28. Edit "leading" to "which can lead" because the cited study is a case study, not an exhaustive, comprehensive study of glaciers writ large.

Line 32. Edit "the glacial and periglacial environment is experiencing" to "both glacial and periglacial environments are experiencing."

Line 33. Edit "The ice avalanches (IAs)" to "Ice avalanches (IAs)."

Line 35. Edit "will pose" to "can pose."

Line 36. Delete "the" that precedes "Ngari, Tibet."

Line 38. The lake reference is unclear. Perhaps "caused a lake dammed by previous IA and debris-flows to collapse" could clarify.

Line 47. Revise to "the conservation of mass and momentum are used to."

Line 49. Edit "approximate" to "approximation."

Line 53. Capitalize "Bowdoin Glacier."

Line 55. Edit "Due to" to "Despite."

Line 56. Revise "the simulation research" to "research simulating temperate glaciers."

Line 59. Change to "This research gap."

Line 62. Change the "will" to "can" in "rapid sliding of the glacier can lead to IAs."

Line 64. Change "different" to "differences."

Line 75. Change to "Midui Glacier."

Line 81. Define firn basin.

Line 94. What penetrates into the glacier? Crevasses?

Line 102. Edit to "days prior to the triggering" for each parenthetical statement.

Line 109. Check significant figures on precipitation. Is it really known to a fraction of a mm?

Line 156. There is not necessarily a large amount of meltwater at the temperate glacier base. Revise to clarify.

Line 240. Correct spelling on "thickness."

Line 241. Revise "section" to "cross-section."

Line 244. Change "and the heat flux can enter and leave the domain," to "and the heat flux is not fixed."

Line 246. Justify setting the wind speed to 1.01 m/s.

Line 282. Change "station" to "stations."

Line 289. Clarify if it is the air temperature, or ice temperature, referenced.

Line 312. Change "material balance" to "mass balance."

Line 316. Change "sun" to "glacier surface."

Line 320. Delete "and the glacier is retreating." Negative mass balance does not instantaneously (or necessarily) result in glacier retreat.

Line 327. Edit section title to "Glacier flow velocity and mass flux"

Line 343. Delete, "indicating that the glacier is melting more and more violently." Ice flow acceleration does not in and of itself indicate enhanced melt.

Line 349. Edit "Glacier stability analysis is to study" to "Here, we conduct glacier stability analysis to study."

Line 351. Define FOS.

Line 371. "Glacier is more prone to collapse." This is an incomplete sentence. Perhaps combine it with the previous sentence.

Line 396. Clarify whether the referenced statistical analysis is in this study or prior study.

Line 407. Change "crevasses expansion" to "crevasse expansion."

Line 428. Change "the multi-physical factors and their coupling mechanisms in the glacier flow process" to "multiple, coupled physical factors in glacier ice flow."

Line 430. Revise. Misleading. The conceptual model was not developed. Instead, models developed during prior studies were applied to the study glacier.

Line 440. Clarify if the stated significance refers to statistical significance.

Line 449. Delete "/."